# The Dual Role of Microplastics in Marine Environment: Sink and Vectors of Pollutants

**Michele Arienzo** [1,*] **, Luciano Ferrara** [2] **and Marco Trifuoggi** [2]

1   Department of Earth Sciences, Environment and Resources, University of Naples Federico II, Via Vicinale Cupa Cintia 21, 80126 Naples, Italy
2   Department of Chemical Sciences, University of Naples Federico II, Via Vicinale Cupa Cintia 21, 80126 Naples, Italy; luciano.ferrara@unina.it (L.F.); marco.trifuoggi@unina.it (M.T.)
*   Correspondence: michele.arienzo@unina.it; Tel.: +39-081-2538166

**Abstract:** This review is a follow-up to a previous review published in Journal of Marine Science and Engineeringon the issues of accumulation, transport, and the effects of microplastics (MPs) in the oceans. The review brings together experimental laboratory, mathematical, and field data on the dual role of MPs as accumulators of hydrophobic persistent organic compounds (POPs), and their release-effect in the marine ecosystem. It also examines the carrier role, besides POPs, of new emerging categories of pollutants, such as pharmaceuticals and personal care products (PPCPs). This role becomes increasingly important and significant as polymers age and surfaces become hydrophilic, increasing toxicity and effects of the new polymer-pollutant associations on marine food webs. It was not the intention to provide too many detailed examples of carriers and co-contaminants, exposed marine species, and effects. Instead, the views of two different schools of thought are reported and summarized: one that emphasizes the risks of transport, exposure, and risk beyond critical thresholds, and another that downplays this view.

**Keywords:** microplastics; oceans; vehiculation; bioaccumulation; effects; persistent organic pollutants; heavy metals; pharmaceuticals; personal care products



## 1. Introduction

The most serious threat to the ecological balance of the world's oceans is the large quantities of plastic that reach seas around the world every year. In 2016 alone, 335 million tons of plastic were produced, of which approximately 10% ended up in the sea [1]. It is forecasted that there could be more plastic than fish in the ocean by 2050 [2]. In the ocean, plastic items undergo a process of mechanical disaggregation, forming microplastics (MPs) <5 mm in length, and nano plastics (NPs) <100 nm, in at least one dimension [3]. Whereas MPs were investigated to a certain extent, NP behavior and toxicology is quite obscure due to the absence of a common standard methodology of measurement. MPs, once ingested by an organism, cannot be decomposed by enzymes and, hence, can lead to fatal outcomes at sub-cellular, cellular, individual, and population levels. This can lead to alteration of gene expression, oxidative damage, antioxidant response, inhibition of cell division, alteration of fatty acid metabolism, reduced growth, reproductive activity, loss of biodiversity, and decline of the population [4]. The response in a single organism can influence the whole community, in terms of nutrient and food web dynamics, biodiversity and community composition, habitat structure, and disease dynamics [4,5]. MPs can be also harmful for the environment being vectors of chemical pollutants. Before spreading to the environment, MPs receive (as part of a chemical fabricating cocktail) chemical additives, which, thus, represent the first vehiculated chemicals that serve to ameliorate their physical and chemical properties. Phthalates, bisphenol (BPA), and polybrominated diphenyl ethers (PBDEs) are used as emollients, flame retardants, and antimicrobials, and are known for being endocrine disruptors and carcinogens [6], disrupting the reproductive system

and biodiversity, and inducing cancers sensitive to hormones [7]. The role of additives and their potential chemical impact after ingestion is an open issue [8]. Besides additive addition, MPs may interact accidentally with xenobiotics in the natural environment, as in wastewater plants, urban runoff, and landfill leachates, moving up to very remote areas by wind and oceans currents. MPs, moving harmful contaminants and pathogens from land to oceans and, subsequently, to humans via feeding routes, are potentially even more toxic [9–17]. MPs may be carriers of titanium dioxide nanoparticles, heavy metals (HMs), POPs, polycyclic aromatic hydrocarbons (PAHs), such as pyrene, fluoranthene chrysene, phenanthrene, polychlorinated biphenyls (PCBs), congeners 18, 31, 138, and 187, polybrominated diphenyl ethers (PBDEs), and dichloro-diphenyl-trichloroethane (DDT)) [18–28]. At present, there is no unique opinion—specifically, many papers doubt a significant MP vehiculation role, and few studies were carried in ocean environmental conditions [29]. Other studies do not doubt vehiculation, but they question whether vehiculated xenobiotics produce bioavailable toxicants upon ingestion [30] and affect reproductive activity, immune response, oxidative stress, cellular toxicity, inflammation, and cancer [31]. There is, in fact, an open debate regarding the transfer of biological pollutants—whether it is considered significant or negligible compared to other routes where pollutants are in direct contact with biota, by direct uptake from water and via ingestion of contaminated preys [32–35]. In regard to humans, some authors consider that exposure to MPs and their vehiculated co-contaminants is low due to limited dietary uptake [35].

This review observes the different studies and outputs on the vehiculation of MPs in oceans, their actual known ability to bind pollutants, the mechanisms of adsorption, the factors that (in the oceans) can positively or negatively alter the interaction, pollutant bioaccumulation, and exposure risks. This review attempts to clarify the open question on whether MPs can be real vectors and transmitters of contaminants exposing organisms to significant ecotoxicological risk.

### 1.1. The Overall Problem of Co-Contaminant Vehiculation

Sorption of pollutants is related to physical and chemical properties of pollutants, and of each plastic, i.e., surface and crystallinity [36], and to their breakdown through photoinduced surface oxidation, physical, chemical, and biotic factors [37]. Even before weathering by solar radiation, MPs present optimal hydrophobic features for chemical sorption of metals, PAHs, PCBs, due to the high surface-to volume ratio and, hence, present high transferability to biota [38,39]. In smaller sizes, their ability to aggregate and transport toxic substances and release them from seawater to organisms increases [40]. Through weathering, MPs become a complex and dynamic mixture of polymers and additives, meaning that various organic and inorganic contaminants can bind to form an "eco-corona", increasing the density and surface charge of the particles and changing their bioavailability and toxicity [41]. Thus, contaminants on MPs are seldom present as a single chemical and usually comprise a complex mixture. Uptake of xenobiotics from such mixtures is poorly understood and questions on whether components of the mixture influence the uptake of other components have not been seriously addressed. Organic and inorganic chemicals can interact with MPs by hydrogen bonds, hydrophobic or electrostatic interactions, and van der Waals forces [42]. Other important parameters concern the molecular weight, and hydrophobicity of the contaminant: the lower they are, the lower it is the mass diffusion [43].

Furthermore, in many marine organisms, gut surfactants could significantly increase the desorption rates of pollutants from eco-corona and, hence, their overall toxicity [36].

Another important regulating sorption factor is represented by the size of the polymer. In fact, nanoplastics (NPs) exchange pollutants at higher rates, with respect to MPs because of higher surfaces and direct proximity to polymers [44]. The sorption rate of nano plastics is reported to be two-fold higher than MPs [43]. As the nano state particles tend to aggregate, this could decrease the available sorption surface; however, this does not occur,

since we have an increase of the surface/volume ratio [43]. It is unknown is if weathered plastics increase sorption through cracking in the amorphous regions or decrease sorption due to crystallinity increase and, hence, more closely packed chains [36]. Furthermore, the aging of MPs favors the formation of surface carbonyl functional groups [37] and, consequently, the adsorption of organic pollutants is not limited to POPs, but also concerns hydrophilic compounds, which have much higher concentrations and are ubiquitous in the environment. The increased sorption capacity of hydrophilic compounds is a matter that deserves more attention, because aged MPs are the fate of all plastic debris. An emerging class of hydrophilic compounds is represented by pharmaceuticals, personal care products (PPCPs), viruses, and pathogens. The effects of sorbed MPs PPCPs on organisms are difficult to predict and little is known about their fate and toxicity. The uptake and distribution of PPCPs onto pristine and weathered plastic pellets, as well as their bioaccumulation in organisms, are not yet well-understood [36]. Once adsorbed, they might be released to the living organisms by direct ingestion or predation along the food chains [24,45,46]. The described xenobiotics are often defined, by the current literature, as sorbed-co-contaminants. Figure 1 shows some of the routes of MP interaction with organic pollutants.

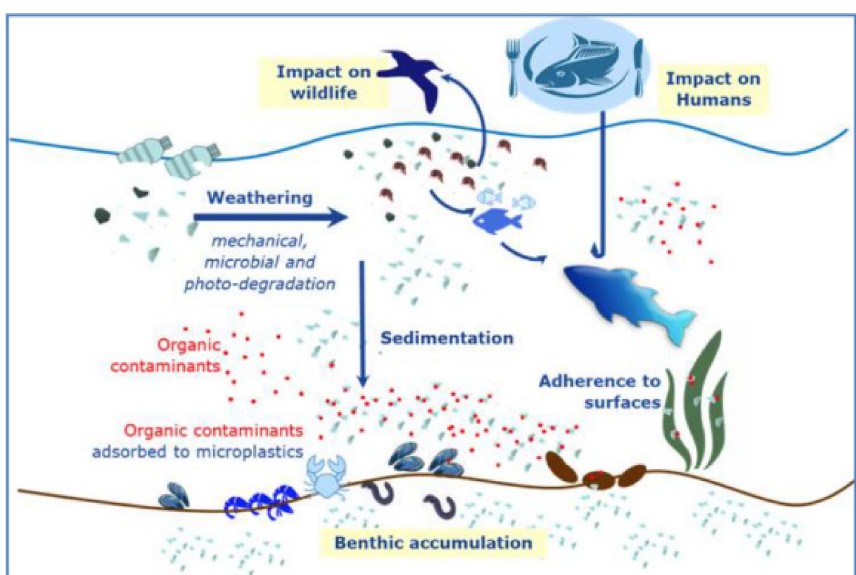

**Figure 1.** The fate of MPs in the oceans and the impact on living organisms. Source: European Commission. Microplastics: Focus on Food and Health.

### 1.2. The Protagonists

#### 1.2.1. MP Nature

The most common synthetic polymers found in seawater are polyethylene (PE), polypropylene (PP), polystyrene (PS), polyvinylchloride (PVC), and polyethylene terephthalate (PET) [46], with larger abundance of PE and PS [22,23]. All of them, as stated above, are mixed with plastic additives, which can serve as flame retardants (bisphenol A (BPA) and alkylphenol); to improve resistance to heat (PBDEs); to prevent damage from oxidation (nonylphenol); to provide resistance to biodegradation (triclosan) [47]; and to increase emollient features (phthalates) [48]. Many of these compounds can cause serious toxicological effects, such as endocrine disrupting chemicals [49]; this is why BPA was banned in several products (i.e., being BPA-free), and replaced by bisphenol S and F, BPS, and BPF. However, these products behave in the same way as BPA in terms of estrogenic, antiestrogenic, androgenic, and antiandrogenic activity [50].

1.2.2. The Potential Sorbing Xenobiotics

The vehiculation of HMs, POPs, and PPCPs by MPs in superficial sediments threatens the world's aquatic resources [51]. It was reported that the concentration of pollutants on MPs are higher than in the enclosing environment due to the high sorption attitude of the fragmented polymers [52]. Affinity, in terms of similar hydrophobicity and extended polymer surfaces, are the two key factors governing this phenomenon. Pellets, for instance, can concentrate on their surfaces, polychlorinated -biphenyls, up to six-fold their concentration in seawater [53]. Pellets with high loads of pollutants were reported in areas near contaminant sources, such as industrial sites and ports [23].

When MPs are in a virgin state, they mainly have hydrophobic surfaces, and interact with POPs, such as hexabromocyclododecane (HDCB), PAHs, PCBs, dichlorodiphenyltrichloroethane (DDT), phenanthrene (Phe) and bis-2-ethylhexyl phthalate (DEHP) [32,54,55], because they possess high, specific surface areas and hydrophobicity. MPs, over time, meet diatoms, hydroids, filamentous algae, and tarry residues; this increases accumulation of pollutants via sorption by particles and accumulation by biofilms [56]. Some authors [54,57] estimated a concentration range of POPs in marine MP pellets of 1–10,000 ng/g. Sorption of DDT, PAHs, hexachlorocyclohexane, and chlorinated benzenes was demonstrated, particularly in laboratory trials [52,58,59]. Moreover, metal adsorption can be consistent [60], especially in aged polymers, since polarity increases, up to 300 µg/g, for Al, Fe, Cu, Pb, and Zn. Adsorption and accumulation of metals come from contact with industrial wastes, the process of combustions, and antifouling paints [60,61]. The current literature displays limited studies on the mechanisms of adsorption of HMs to MPs [61,62]. A study by Ashton et al. [63] reports HMs adsorption onto polyethylene pellets. A later study by Holmes et al. [60] evinces that sorption equilibrium is reached in 25–100 h and that increased weathering and interaction with organic matter increases the sorption due to larger surface area and generation of surface anionic active sites. The sorption mechanisms of metals on MPs remains rather unclear, and it seems that sorption is driven more significantly by the rate of aging rather than the kind of polymer [64]. The same conclusion is formulated by Brennecke et al. [60], reporting how PVC and PS can accumulate metals leached from antifouling paints into the seawater, up to 800-fold, than levels in the surrounding sea water. As stated above, plastics, besides POPs and metals, can also accumulate PPCPs, including >4000 products worldwide. It is estimated that up to 15,000 tons of antibiotics are released annually into the European environment and have a high possibility of interacting with aged MPs, because of their hydrophilic, oxygen-containing functional groups. Three are the classes of antibiotics very common in coastal water, sulfonamides (SAs), quinolones (QNs), and macrolides (MLs), including erythromycin, oleandomycin, and spiramycin. The most diffused antibiotics in oceans [37] are oxytetracycline, tetracycline, sulfamethoxazole, ciprofloxacin, and trimethoprim, which are endocrine disruptors, and impact crustaceans and fish, alter the levels of hormones in the blood, and can induce cardiovascular collapse [65]. In European coastal superficial water, the most common antibiotics are erythromycin, amoxicillin, ciprofloxacin, sulfadiazine, and clarithromycin [66]. One hundred thirteen pharmaceuticals and pharmaceuticals metabolites were detected in coastal waters [5]. Figure 2 shows the vehiculation of endocrine disruptors by plastics in humans. Marine and offshore environments can be impacted, in their waters and sediments, by the presence of many groups of antibiotics from aquaculture, land farming, and wastewater [51]. Antibiotics are feared in all types of water, because (i) they induce resistance; (ii) resistance genes transfer horizontally between microbial populations, and (iii) antibiotics can have direct wide-spectrum toxicity to organisms [67].

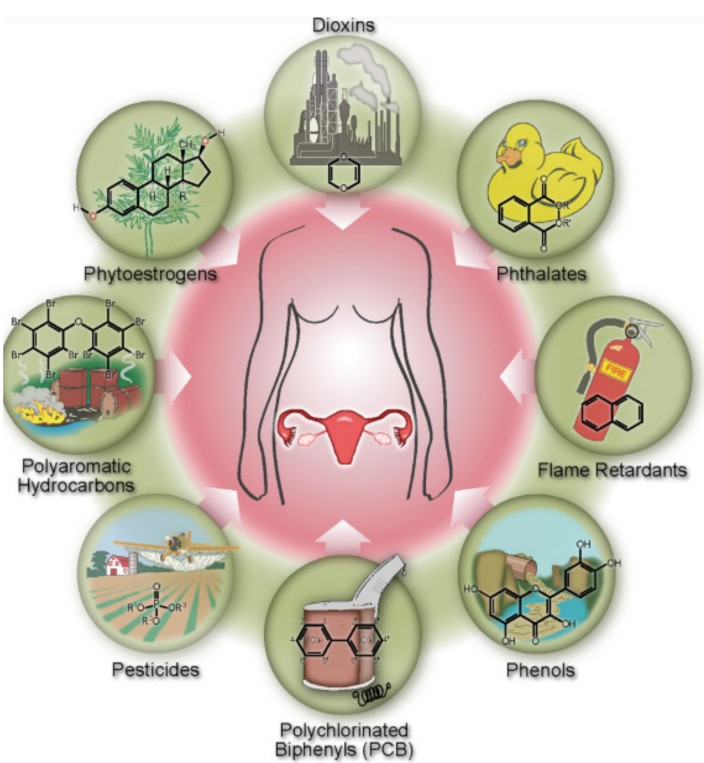

**Figure 2.** Vehiculation of endocrine disruptors in humans by MPs. Source: *PLoS ONE*.

Cosmetics as UV filter preservatives, parabens, triclosan, are emerging pollutants of particular concern. These products are continuously released into the aquatic environment and their ecological concern is due to the large volumes produced, their persistence, bioactivity, bioaccumulation capacity [68], and to the scant removal efficiency of wastewater treatment facilities [69–71]. Cosmetics, with respect to pharmaceuticals, are produced in much higher quantities, do not go through metabolic transformation, and reach the seawater unaltered in great amounts during washing, showering, and bathing [72]. While the ecotoxicological behavior of surfactants, largely present in PCPS and discharged in water ways, is well-known [73]; that of UV filters, parabens, and triclosan as emerging environmental contaminants is still unclear [68].

## 2. Sorption of Pollutants

### 2.1. The Two Sorption Routes

MP sorption of pollutants are governed by the kind of polymer, color, dimension, degree of aging, and the conditions of the sorbing media, such as pH, salinity, and temperature of the sea water [74]. Pollutants can interact with MPs through two routes: (i) absorption by the weak van der Waals forces, where partition is regulated by the octanol to water coefficients, and pollutant molecules are in a soluble state nearby the sorbent surfaces; (ii) adsorption, involving a more complex set of forces besides van der Waals, i.e., ionic, steric, π–π interactions, and covalent bonds [40,75,76].

The two routes depend on several factors, and the dominant route depends on the concentration of the pollutant in the sorbing media: if this is low, the adsorption prevails with several interacting forces acting on polymer surfaces; if high, the absorption is dominant, due to the much larger volume to accommodate the molecules [40,77,78]. Besides the pollutant load, the features of the polymer and the POPs are also important [20]. Sorption studies of pollutants in marine environments, especially in regards to POPs. In such environments, POPs, being high hydrophobic and lipophilic, interact with several non-polar surfaces, such as MPs, and suspended organic material and sediments [79]. The interaction is driven by equilibrium partition ratios related to octanol to water partition ratios [75,80] and diffusive mass molecular diffusion [40]. There is a positive correlation with

hydrophobicity [52,81], expressed by the octanol/water partitioning coefficients (Kow). Thus, the lipophilicity and hydrophobicity of MPs are fundamental in accumulating POPs as well the high surfaces of the polymers, with respect to their volume [53]. Due to these features, plastics can accumulate POPs up to two-fold when compared to natural sediments and soil, and up to six-fold the levels in seawaters [53,82]. Once hydrophobically sorbed onto MPs, POPs can be desorbed, depending on the sorption equilibria of the media in the marine system; both processes are through diffusion along concentration gradients [83].

### 2.2. The Polymer Factor of Sorption

Several factors regulate the sorption of co-contaminants, such as size, shape, color, crystallinity, additives, polymer types, the chemistry of their surfaces, and their effects in organisms [4]. However, more than others, the uptake of contaminants by MPs is driven essentially by the type of polymer. This is because each polymer has its own specific surface area and crystallinity [84,85].

As displayed in Figure 3, density, crystallinity, size, color, polymer, age, and molecular weight represent the most important features characterizing polymer sorption properties of POPs [24,40,86]. The variability of their morphology (spheres, fibers, foams, fragments, and beads), as well of density, 16–2200 kg/m$^3$ [87], makes the interaction processes among the most varied and polymer specific. MPs of darker color can have more significant sink of DDT, PAHs, PCBs, as reported by Frias et al. [22], hypothesizing the concomitant occurrence of physical and chemical sorption.

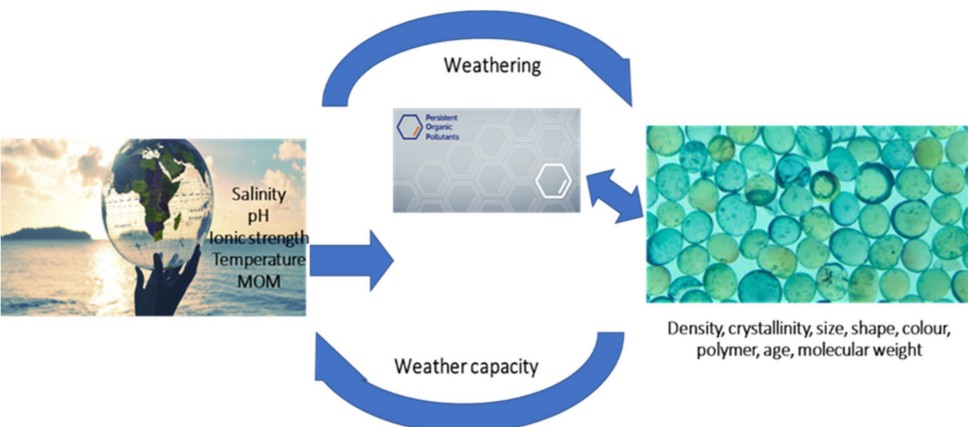

**Figure 3.** Factor governing the interaction between MPs and POPs.

If the hydrophobicity of adsorbate has a key role in the sorption process, we do not have to ignore that polymers display different structures with crystalline and amorphous distinct moieties [79]. The amorphous region is made by long molecular chains, forming irregular and disordered entangled coils, whereas the crystalline area is made by chains, rearranging upon freezing and formation of ordered regions, with a typical size of the order being 1 micrometer [88].

A Deeper Look at the Polymer Structure

Figure 4 display the model of a spherulite with localization of the amorphous region. Polymers crystallize from melting by three routes: cooling, mechanical stretching, and solvent evaporation. The optical, mechanical, thermal, and chemical properties of semicrystalline polymers are determined by the degree of crystallinity and size and orientation of the molecular chains. The crystalline areas involve the presence of ordered regions with alignment of the molecular chains, folding together, and forming ordered lamellae [89]. Inside the crystalline region, chains are only rarely aligned parallel, and there is coexistence of aligned and folded chains, giving rise to semicrystalline structures. The degree of crystallinity ranges between 10 and 80%, and crystallized polymers are named "semicrystalline".

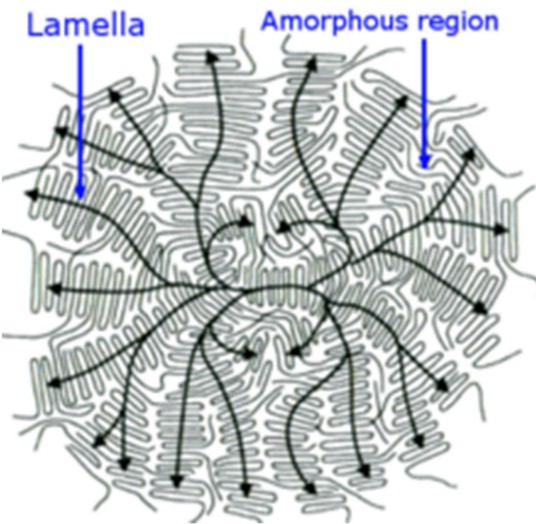

**Figure 4.** Schematic model of a spherulite with the black arrows indicating the direction of molecular alignment.

Examples of semi-crystalline polymers are linear polyethylene (PE), polyethylene terephthalate (PET), polytetrafluoroethylene (PTFE), and isotactic polypropylene (PP) [90].

The chains interact mainly by van der Waals forces, whose intensity depends on the distance between the parallel chain segments [90]. Crystalline areas are generally more densely packed than amorphous areas. This results in a higher density, up to 15%, depending on the material. For example, polyamide 6 (nylon) has crystalline density $\rho_c$ = 1.24 g/cm$^3$ and amorphous density $\rho_a$ = 1.08 g/cm$^3$ [90]. Most methods of evaluating the degree of crystallinity assume a mixture of perfect crystalline and totally disordered areas; the transition areas are expected to amount to several percent. These methods include density measurement, differential scanning calorimetry (DSC), X-ray diffraction (XRD), infrared spectroscopy, and nuclear magnetic resonance (NMR). The measured value depends on the method used, which is therefore quoted together with the degree of crystallinity [90]. In addition to the above methods, the distribution of crystalline and amorphous regions can be visualized with microscopic techniques, such as polarized light microscopy and transmission electron microscopy.

X-ray diffraction can be used to evaluate the regular arrangement of atoms and molecules producing sharp diffraction peaks, whereas amorphous regions result in broad halos. The diffraction pattern of polymers usually contains a combination of both. The degree of crystallinity can be estimated by integrating the relative intensities of the peaks and halos [91]. An additional technique to evaluate the crystallinity is represented by infrared spectroscopy (IR): infrared absorption or reflection spectra from crystalline polymers contain additional peaks, which are absent in amorphous materials with the same composition. These signals may originate from deformation vibrations of the regular arrangement of molecular chains. From the analysis of these bands, the degree of crystallinity can be estimated [91]. Below their glass transition temperature (Tg), amorphous polymers are usually hard and brittle because of the low mobility of their molecules. Increasing the temperature induces molecular motion, resulting in the typical rubber elastic properties.

The degree of crystallinity influences their mechanical properties: higher crystallinity results in a harder and more thermally stable, but also more brittle, material, whereas the amorphous regions provide certain elasticity and impact resistance [92]. These polymer properties make pollutants have more access inside the amorphous regions, where chains are disordered aligned, and with empty voids and free volume increasing diffusion of hydrophobic chemicals between polymeric chains. Hydrophobic chemicals can hardly access high crystalline polymers, such as PS, and easily diffuse into low crystallinity polymers, such as PP and PE [79].

The above considerations mean that important polymer features governing sorption are the temperature of glass transition (Tg), the degree of crystallinity, and of van der Waals forces. When temperature is below Tg, polymer chains have no steric mobility, whereas when it is above Tg, chains are flexible and steric mobility is higher [93]. PS represents a good example of high crystalline polymer, which for the above considerations has very low sorption power of POPs, whereas PE and PP are low crystalline polymers and, hence, good POP sorbents [24]. However, crystallinity is not sufficient to drive sorption alone, since size, i.e., surface area and free volume features strengthen PE–POPs sorbent power [24,76].

One additional important factor is the discrimination inside the amorphous moiety of a glassy or rubbery region, depending on the Tg. If it is glassy, as PVC and PS, the chains are more cross linked and condensed, are not crystallizable, and represent strong adsorption sites for POPs [40,81,86]. PE and PP are instead rubbery polymers where permeability and diffusion is significant and, hence, absorption is dominant [81]. Thus, rubbery polymers, PE and PP, were found to represent a more important sink of POPS in the oceans, relative to glassy PET and PVC, with faster kinetic of sorption [24]; most authors agree to define PE as the greatest POP concentrator in absolute [20,76]. Each polymer has its own specific sorption behavior. In this regards, Yu et al. [36] reported data from 12 papers on MPs sorption of metals, naphthalene, phenanthrene, pyrene, pesticides, dichlorodiphenyltrichloroethane, PCBs, 3,3′,4,4′-tetrachlorobiphenyl, 2,4,4′-trichlorobiphenyl 2,4′,5-trichlorobiphenyl, antibiotics, oxytetracycline, tetracycline, sulfamethoxazole, ciprofloxacin, trimethoprim, and reported the sorption parameters of the linear, Langmuir, and Freundlich models. It is interesting to note that PE and PS were the most adsorbing polymers. The high surface area and free volume are responsible for the strong PE sorption of hydrophobic compounds [94], whereas the key dominant sorption factor for PS is that polymer filaments are more spaced by the presence of aromatic rings, allowing easy diffusion of xenobiotics [84]. However, this is not always true due to specific polymer–xenobiotic specific interaction, as is the case of that between trimethoprim and polyamide [95]. A comprehensive review by O' Connor et al. [80] deals with the partitioning of POPs with Kow in the range of 2–8, into different plastics, in a consistent number of field and laboratory assessments. The main output of the work is that high density PE (HDPE) and low-density PE (LDPE) behave similarly and have higher partition than PP, PVC, and PS. It seems that sorption is related to the extension of the amorphous moiety, where chains are not so ordered, and surfaces are not so closed and, hence, a certain free volume is available for chemical diffusion. Thus, this area is the location where the chains are flexible and can become more relaxed, depending on the environmental conditions. We can synthesize the xenobiotic sorption process, as the adsorption when sorption occurs on the external surfaces of polymers, and absorption as sorption diffusion in the bulk of the polymer. The extent of adsorption and absorption depends on the nature of the xenobiotics, in terms of polarity and molecular weight, and of the polymer, in terms of rate of crystallinity and phase morphology.

### 2.3. The Weathering Factor

Polymers can oxidize by thermal, photochemical, and biological processes, with a modification of polymer features [96]. However, the solar radiation is, especially, the most important aging factor. Plastics once exposed to solar irradiation are converted in MPs and NPs by action of UV rays on the unsaturated structures of polymers and autocatalytic oxidation on the surface of the exposed particles. The UV photo radiation causes a sequence of chemical reactions involving the formation of polymeric radicals, oxygen addition, and loss of hydrogen, leading to slit of the polymer chains [86]. In detail, when MPs adsorb UV radiation, there is generation of free radicals due to breaking of C−H bonds inside the chains of the polymers [97]. The free radicals react with oxygen and produce hydroperoxides [98]. Therefore, ultimately, the aging enriches the surfaces of carbonyl and hydroxide radicals, and the properties of the pristine MPs change toward oxidation state conditions. There is evidence [38] that, with aging, the surfaces of pristine MPs enrich carbonyl and hydroxide radicals, and present small wrinkles during aging,

and the amorphous polymer crack opening its structure [99,100]. The produced cracks on the aged MPs cause the formation of additional sorption sites and increase retention of pollutants. Plastic debris in the oceans may meet two different environments; one well aerated and irradiated, eliciting fragmentation processes, and one where the cold and anoxic environment slow down the degradation process [28,101].

The weathering process modifies the polarity and, consequently, sorption features of the polymer particles, which become less crystalline, and more amorphous [19]. On the surface of the polymers, new functional groups form as ketones, esters, carboxylates, and hydroxyl groups and, hence, the sorption surfaces shift from hydrophobic to hydrophilic, reducing the potential retention of POPs [53]. On the other hand, POPs sorption can increase, since, under oxidizing conditions, there is a separation of the chains, a rupture of sites at the amorphous zone, and fragmentation in smaller particles, with higher flexibility, lower molecular weight, and predominance of amorphous moiety available for POPs diffusion [102,103].

Thus, the weathered surfaces can increase or decrease the interaction with the POPs depending on their polarity, Kow, diffusion/penetration capacity, and on the adaptation of the newly formed open surfaces and the morphology of the specific polymer [104]. The above exposed effects of aging reveal that weathering makes polymers capable of interacting with PPCPs in the one hand, and increases or decreases POPs retention on the other hand.

Of course, not all polymers have the same resistance to weathering, depending on their intrinsic characteristics and on the addition of additives and stabilizers [86]. In conclusion, some studies retain that the main induced effect of weathering is due to the increased crystallinity and, hence, the decreased POPs sorption [52]. Other studies attribute cracking, molecular weight fragmentation, formation of new available areas, increased pore size, and volume to increased POPs sorption [22,79]. Moreover, increased exposure times and biofouling increases interaction with POPs [19,76].

A later study of Fisner et al. [105] correlates the phenomenon of sorption with aging and with the increasingly darker colors of MPs. The same authors reported higher sink of high molecular weight PAHs by darker pellets. Other authors, [93], studied the effect of aging on the MPs sorption of POPs (benzene, toluene, ethyl benzene and xylene (BTEX), and four tertiary butyl ethers). The sorption on PP and PS appeared correlated with Kow, and it was significant only for BTEX. While aging had no effect on PP-BTEX sorption, on PS it significantly decreased. Karapanagioti and Klontza [84] reported higher phenanthrene sorption on eroded PP pellets; Geodecke et al. [106] reported a higher load of difenoconazole on aged PP. This shows how there is no unique relationship among sorption and aging, color, and crystallinity increase. Müller et al. [93] noted decreasing PS molar mass and a deterioration of polymer chains by aging and, hence, greater flexibility of chains, representing discontinuity points and potential new adsorption sites for POPs onto the glassy polymer. An increase in diffusiveness and adsorption capacity due to polymer cracks did not occur, even if it was expected. The authors concluded that hydrophobicity of the sorbates was the driving MPs sorption factor.

Since all plastics become hydrophilic over time, they can become "sink" and vectors of PPCPs. In this regard, Liu et al. [37] studied the sorption of ciprofloxacin by UV catalyzed weathered PS and PVC, and reported that sorption increased with the following order: PS < PVC < aged PVC < aged PS. The sorption ability of weathered PS and PVC were more than 120 and 20% higher than the unaged polymers [37]. The authors discovered how weathering significantly increased the adsorption of ciprofloxacin and that interaction occurred by electrostatic interaction, cation exchange, hydrogen bonding, and π–π interaction.

### 2.4. The Environmental Factors

There are other factors strictly related to MP features that govern POPs sorption, i.e., salinity, temperature, pH, and content of organic matter of the environment [46]. Salinity

increases the agglomeration of MPs and, hence, the number of sites for POPs sorption [43] and modifies the water solubility of the contaminant [80]. Even in regards to hydrophilic compound, salinity has an opposite effect. Liu et al. [37] observed that salinity increases the density of the solution and reduces the mass transfer of ciprofloxacin (CIP) to the polymer phase of pristine and aged PVC and PS. This is because $Cl^-$, in the case of PVC, increases the cohesive density between the polymer chains, reducing the free volume for polymer diffusion [94]. Whereas $Na^+$, in the case of PS, reduces the cation exchange mechanism, as $Na^+$ saturating the polymer surface impedes any electrostatic interaction, being ciprofloxacin positively charged at pH < 6.0.

Other studies pointed out the role of temperature, pH, and load of organic matter. In general, high temperature and low pH enhance release of POPs [46] and high loads of organic matter decrease MPs sorption by competitive mechanisms [78]. Some reports reveal an increase of sorption with the temperature above 15 °C, and up to an optimal value of 25 °C, outlining an endothermic sorption [107]. Liu et al. [37] studied the interaction of PS and PVC with ciprofloxacin, which has a pKa1 = 6.1 and, hence, the molecular structure and speciation change under different pH conditions (cation $CIP^+$ at pH below 5.0, anion $CIP^-$, at pH above 10.0, and zwitterionic $CIP^0$ at pH 7.0 [108,109]. They showed that when the pH was below 6.0 it was the electrostatic interaction that governed the interaction between the negative charge on the PS and PVC surfaces and the positive charge of the pollutant with a maximum adsorption at pH 5.0. At low pH, pristine PS, with respect to PVC, can also add an additional strengthening interaction mechanism, i.e., π–π bonds. When pH increases >9.0, adsorption decreases due to electrostatic repulsion between the antibiotic, which becomes anionic, and the oxygen groups enrich the aged MPs [37].

## 3. The Pollutant Bioavailability

The presence of xenobiotics, POPs or PPCPs, on MP surface represents an additional risk for biota. Once plastic particles are assumed through diet, they can cause mechanical blockage of the gastrointestinal tract and toxic damage from xenobiotics stripping [5,110]. Pollutants may accumulate in the fat tissues or from the digestive apparatus can pass to the circulatory system, causing serious toxic threats to vital organs [19].

Biota can adsorb the transported MP pollutant by direct contact with the skin, skeleton, gill, and gut [19], or by indirect contact with the desorbed pollutant in the liquid phase i.e., water, and internal fluids of the biota [40]. Of course, desorption is much more intense in the digestive apparatus because of the combined action of acids, organic substances, and surfactants [111,112]. Thus, POPs can be released by MP vectors once they are inside the organism, externally in the water phase, or into food/prey, and then taken by the organism, with processes influenced by pH, gut surfactants, temperature, and time of exposure [30,32]. In the case of HMs, Holmes [64] reports higher concentrations of metals on plastics than in water. The same authors reported that adsorbed metals are bioavailable by extraction in the digestive tract, being an acid environment. Uptake via pore water or internal fluids means that POPs need to be desorbed, with desorption affected mainly by partition and diffusion. The nature, kind, and concentration of gut fluids and distance of contact influence mass diffusion transfer [40]. In the external environment, driving factors of desorption are water, dissolved organic matter, bacteria, diatoms, black carbon, and natural fibers [30].

The exposure of marine biota to MPs and their co-contaminants may affect different trophic levels, depending on intrinsic MPs features, such as density, size, and structure and, hence, exposure may regard water surface, water column, or sediment and, hence, planktonic, or benthonic organisms [22]. The most exposed organisms are filter feeding and benthonic organisms with a high bioaccumulation of power, which can thus represent a serious threat to upper lower trophic levels [59]. There are still no studies reporting damage to human health resulting from the consumption of foods where MPs and co-contaminants have bioaccumulated. Another unclear aspect is if there exists any interactive effect, i.e., if there are synergistic effects of the polymer-contaminant combination on ecotoxicology [83]. An early literature study [24,113] reported on the bioavailability of adsorbed pollutants

to organisms, which can be chronologically cross linked with early studies, reporting superficial sorption of these contaminants on polymers [19,21,53].

Bioavailability of sorbed pollutants is significant in organisms of higher trophic levels due to release from the gut, depending on high blood temperature, lower pH, and presence of surfactants [46]. Interestingly, in this regard, is the example reported by Batel et al. [114] on the desorption of benzo(a)pyrene from PE in the digestive system of zebrafish, as revealed by increased biomarker activity of cytochrome P450 11A in the liver. A similar trophic transfer for MP adsorbed fluoranthene was observed in the chain copepods–larval kill fish, and from fathead minnows to striped bass [115]. Many studies—to evaluate the bioavailability of sorbed POPs on MPs—are carried out under determined laboratory or pilot conditions that attempt to simulate real field scenarios, but they cannot obviously be very realistic. For example, let us look at the study by Paul-Pont et al. [116] on the bioavailability of fluoranthene from PS to *Mytilus* spp. after 7 day of exposure and 7 days of depuration. The authors reported a higher bioaccumulation of fluoranthene after depuration in mussels exposed to PS-fluoranthene than in organisms exposed to the pollutant alone. This was attributed to the direct effect of MPs on the detoxification mechanism as suggested by downregulation of a glycoprotein involved in pollutant excretion, as well impairment of the filtration activity or presence of beds in the gut [116]. The authors also highlighted direct toxic effects at tissue, cellular, and molecular levels of MPs, together with fluoranthene. In other cases [117], it was shown that POPs, such as phenanthrene and fluoranthene, display different impacts on marine organisms, e.g., copepods—if they are ingested in freely dissolved water form or in an adsorbed PE or PS form, the freely dissolved form being the most impacting.

In the case of PPCP exposure studies, biomarkers are often used to evaluate biochemical, molecular, physiological, and morphological changes [118]. The major difficulty of using biomarkers is the scant possibility of scaling-up to organism and ecological levels, which do not work for all stages of development [5]. Although PPCPs have low values of octanol–water partition coefficients [119], bioaccumulation in fish and molluscs were reported [119]. Bioaccumulation of UV filters, synthetic mosses, and methyl triclosan were reported in marine organisms [119], with degrees of bioaccumulation depending on the pH of the environment and the phylum of the taxa. Some cosmetics, such as UV creams, may have a bioaccumulation factor >5000 in fish identical to that of PCBs [120].

## 4. Interactive Toxicity

The toxicological mechanism of MPs on marine biota are quite numerous and concern oxidative stress via free radical production [121] alteration of immunological responses [121], and of gene expression profiles, reproductive toxicity, developmental neurotoxicity, modification of phenotypes of offspring generations, abnormalities of the external genitalia, and endocrine activities in fish and mammals [37,122–124]. These toxicological consequences add to the major plastic effects of blockages of digestive and intestinal tracts, conducing to suffocation of most marine vertebrates [125], or can cross the gastro-intestinal membranes and accumulate in different tissues by an endocytosis-like mechanism [1].

One of the first evidence reporting a correlation between MP concentrations and PCB accumulation was shown by Besseling et al. [10] in *Arenicola marina,* with a reduction of weight and fitness.

An enhanced toxicological effect was also reported by Kim et al. [126] who investigated the toxicities of variables and fixed combinations of two types of MPs, one of them coated with a carboxyl group, PS-COOH, and one alone, PS, with the heavy metal nickel, Ni, on *Daphnia magna*. They found that toxicity of Ni in combination with either of the two MPs differed from that of Ni alone. They observed that immobilization of *D. magna* exposed to Ni, combined with PS-COOH, was higher than that of *D. magna* exposed to Ni, combined with PS. The results indicate that the toxic effects of MPs and pollutants may vary depending on the specific properties of the pollutant and microplastic functional groups. The authors hypothesized that the higher toxicity of PS-COOH was attributable

to the higher electrostatic interaction between Ni cation and the negative charge functional group and, hence, to higher *Daphnia* Ni ingestion. The findings highlight that the MPs-co-contaminant combinations may have synergistic consequences due to the specific characteristics of the MPs and of the pollutants, and on the presence on MP surfaces of specific functional groups.

Alteration of key physiological processes in the liver and brain of zebrafish was reported by the interactive effects of MPs with sorbed PCBs, perfluorinated compounds, or methyl mercury. Brandts et al. [124] exposed mussels to PS, plus carbamazepine and observed significant alteration of gene expressions, DNA integrity, cellular oxidative stress, and alteration of endocrine systems, compared to individual chemicals [124]. In this latter study, the authors exposed, for 96 h, nanoparticles of PS to Cbz alone and in a mixture of PS + Cbz, and determined the molecular and biochemical biomarkers in the digestive glands, gills, and hemolymph. Other interactive effects were shown by Vijver et al. [127], who exposed *Daphnia magna* to MPs alone and to MPs–dimethoate, an organophosphate insecticide with a low log Kow, and deltamethrin, a pyrethroid insecticide with a high log Kow. *Daphnia magna* survival was not affected by exposure to MPs alone, whereas the effect was visible in the MPs–dimethoate or deltamethrin combination. In other cases, the interaction MPs–pollutant decreased the toxicity, as was observed by Zhang et al. [128]. They exposed the alga *Microcystis aeruginosa* to a combination of MPs–glyphosate and observed no effect on alga growth with MPs alone, whereas toxicity was determined in the case of exposure to glyphosate alone. In the presence of MPs–glyphosate combinations, the strong adsorption affinity of glyphosate for the microparticles reduced the overall inhibitory effect of the herbicide.

Other authors retain that the release of chemicals from MPs is of low importance and has very limited effect on marine biota [123].

## 5. The Two Conceptual Schools

There is a debate questioning whether MPs are of a significant sink and act as vectors that pollute the marine environment. The current literature does not clarify this. There are, in fact, two schools of thought. The first is based on evidence by Teuten in 2009 [20], Avio in 2015 [59], and later, by Batel in 2018 [129]. The authors, using enzymatic, tissue and fluorescence tests [79], reported bioaccumulation of POPs from MP contact or release in tissues of marine organisms. The authors documented xenobiotic toxicity by evaluating the response of the immune system, oxidative generated stress, and genotoxicity damages [59]. Moreover, the vector role may be potentiated by photooxidation aging and concomitant interaction with organic matter, biofilm [53,60,61,130], by the increasing loads of MPs in the oceans [131], and the presence of many MP hot spots.

The second "school of thought", besides the high POP affinity for plastics [84], downplayed the carrying and bioaccumulating of POPs role in the oceans [44,46,132]. Authors [133] state that POP bioaccumulation from a direct exposure pathway or release is very low since MPs represent <3% of the diet. Studies [134–136] reported the absence of any bioaccumulation of POPs as well toxicological consequences.

There is some research [32,74,133] that retain the negligible vector role of MPs and conclude that bioaccumulation can be significant only in locations where high loads of MPs exist and, hence, there is a chance of possible bioaccumulation [74]. Hartmann et al. [40] consider that the bioaccumulating contaminant role of MPs can be significant only in areas where MP volumes and densities are high and there is chance of direct exposure. Ziccardi et al. [74] report that chemicals can concentrate on MPs many folds higher than in seawater, and that the exposure routes from MPs are difficult to discriminate and isolate. It might occur by many other alternative routes as water, sediment, and food. Thus, the ecotoxicological role of MPs as vectors of bioavailable toxic substances [24,43,137] has often been challenged, because MPs are considered secondary vectors of POPs, with respect to the more abundant presence of suspended organic material and natural prey [30].

One critical consideration is that only exposure by biomarkers determination, and not effects, is often tested [73]. The latter authors report that most laboratory studies are carried under maximum gradients of chemical activity. In this unrealistic situation, it is very likely to observe significant POP transference. In real field scenarios, these gradients are utopistic due to the presence of a more complex environment with different compartments and organisms, lower concentrations of MPs and chemicals and, hence, chemical activity gradients will be much smaller [73]. Another (difficult to properly evaluate) POP effect is that it would be very probable that concentration of POPs in plastics, organisms, water, and sediments will be quite different in bench and field scales, with prevailing disequilibrium conditions in field scenarios.

Supporters of the first school of thought base their conclusions mainly on laboratory studies [19,138], where mechanisms, dynamics, and biological effects are assayed [52,81,139]. Even though lab trails present the advantages of mimic and control, the environmental conditions they represent only limit time scale exposure scenarios. Thus, it would be preferable to perform, parallel, field and laboratory trials.

The second school of thought is also supported by mechanistic studies. Research explored the principles governing the MPs–POPs interaction and the parameters regulating sorption/desorption, i.e., MPs weathering rate and concentration, gut surfactants, pH, and temperature [30,32]. Bakir et al. [32] modeled the transfer of POPs from PVC and PE to organisms under various environmental conditions and concluded that their model ascertained food and water as a major combined route of exposure, with a negligible input from MPs.

Koelmans et al. [30] adopted new models and concluded that the bioaccumulation of POPs from natural carriers, such as water, suspended organic particulate, natural diet, and prey, having a partition coefficient like those of MPs [53], is more important. The latter author and Koelmans et al. [30] retain that high hydrophobic chemical tend to concentrate more in water, 99%, DOC and colloids, 0.4%, rather than in MPs. They argue that this can be true because natural carrier loads are higher than those of plastics. Koelmans et al. [30] consider that bioaccumulation and biomagnification can represent a risk only if the threshold effect is exceeded. In a similar way, Bakir et al. [32] doubt if ingested MP-bound pollutants can be transferred at a harmful level as well as the relative weight of MP dietary assimilation and other pathways of exposure [132].

The definition of appropriate threshold effects is complicated by the fact that MPs can be concomitant accumulators of additives and may act as reductants of pollutant concentrations, having opposite fugaciousness gradients between MPs and lipids [44,132].

The poor role of MPs as vectors of POPs [34] appears to be supported by the mass discrepancy among that of plastics, water, and organic substances. According to Gouin et al. [132] and Koelmans et al. [30], the water mass exceeded that of MPs by a factor of $10^{13}$ and the volume of natural organic carbon in coastal areas greater than $10^7$. However, MP estimation can be underestimated due to the lack of standardized MP sampling procedures [140].

Our opinion is that conclusions of the two schools of thought are questionable, and a certain prudence is mandatory; moreover, we do not have to forget that plastics can concentrate POPs by factors up to $10^7$ from water [30]. There is also a lack of unambiguous methods of investigation on the interaction of MPs with biota, based on exaggerated concentrations of MPs, not reflecting natural scenarios. Current research considers a limited number of xenobiotics and polymers of environmental conditions and target organisms. It is also difficult to assess both bioaccumulation and toxicological effects, especially in field situations, due to the small size of MPs and NPs, which are difficult to identify in tissues, even by fluorescent traceability methods [141].

## 6. Conclusions

Studies on MPs as vectors of pollutants are quite few. There is also limited understanding of the toxicity thresholds and translocation in biota by direct internal and external contact. This will be possible only when the current techniques of quantification and

localization, fluorescence microscopy, and electron microscopy ion mass spectrometry of micro and nano plastics, improve.

In general, the evaluation of the vehiculation role of POPs by MPs is performed using thermodynamic mechanistic models, assuming equilibrium within the time of the experiment. However, many observations reveal that this is not often the case and, hence, bioavailability can often be underestimated [115]. It will be very important to conduct further studies on the mechanisms of interaction between MPs and pollutants, the functional groups involved, and toxicity variations.

## 7. Final Considerations and Suggestions

We imagine that readers of this review will want to know what our views are on the role of MP as a sink or vector of pollutants in the oceans.

Our idea is that, at current levels of MPs, the effects are not so dramatic. Of course, many predictions estimate that large quantities of plastic will be released into the sea soon, exceeding that of fish. Perhaps MP levels, which might form a new category of synthetic sediments, will bioaccumulate above critical thresholds and synergistically combine to enhance toxic effects in biota.

We believe that sophisticated mathematical models simulating the most complicated situations in the real environment are often limited, since in the real marine environment, other competitors and numerous factors interfere with the balance or imbalance of sorption processes.

Perhaps, in a few decades, when the pessimistic predictions of MP accumulation will come true, and the seabed is covered by a new layer of synthetic polymer sediment, the weights, and outputs of the two schools of thought will be reconsidered and elucidated. Transformation of current and future levels of MPs into nano plastics will increase the level of ecotoxicological concern, reducing transportability and bioaccumulation of POPs, and increase that of hydrophilic xenobiotics, where the light of research on ecotoxicological risks is still very dim.

**Author Contributions:** Writing-original draft preparation, M.A.; writing—review and editing, L.F., writing-review and editing, M.T. All authors have read and agreed to the published version of the manuscript.

**Funding:** This research received no external funding.

**Institutional Review Board Statement:** Not applicable.

**Informed Consent Statement:** Not applicable.

**Data Availability Statement:** Not applicable.

**Conflicts of Interest:** The authors declare no conflict of interest.

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
