# Peer review of "The Dual Role of Microplastics in Marine Environment: Sink and Vectors of Pollutants"

_jmse, doi:10.3390/jmse9060642_

Round 1

Reviewer 1 Report

The manuscript presents the literature review of the dual identity of microplastics in marine environment, as sink and vectors of pollutants. Authors made a good quality, comprehensive review of the published data, providing their own view of two diverse opinions related to the microplastic behavior in the marine environment. 

I just noticed few minor mistakes like that in line 165 where comma is missing between Fe, Cu and Pb. Please, go through whole manuscript and check for this kind of mistakes.

Besides, I can recommend this manuscript for publication.

Author Response

  • I just noticed few minor mistakes like that in line 165 where comma is missing between Fe, Cu and Pb. Please, go through whole manuscript and check for this kind of mistakes.
  • Answer: we amended the text as suggested for missing comas.

Reviewer 2 Report

  1. Use tables on the mian description items, such as positive and negative effects, dual identity, and so on, because this article is the review paper. Tables with the well-classified contents will be helpful for the bettr understanding of your article.
  2. Check the inconsistent words useed in your article. (Line 33 NPS --> NPs, ageing(line 173) or aging(line 213), Line 182 MIs --> MLs, Line 387 Kow (line 403?), Line 648 time spam? --> time span.
  3. Sec.7. Our opinion -->It is not suitable fot these scientific articles. Out suggestion?

Author Response

  • Use tables on the mian description items, such as positive and negative effects, dual identity, and so on, because this article is the review paper. Tables with the well-classified contents will be helpful for the bettr understanding of your article.

Answer: we agree with the reviewer that the use of tables can be useful in organising the different argument included in the review. However, perhaps given the division of the paper into many sections and subsections with clear indications of the relevant topics covered, including extra tables might be perhaps avoided even to avoid rewriting large parts of the text.

  • Check the inconsistent words used in your article. (Line 33 NPS --> NPs, ageing(line 173) or aging(line 213), Line 182 MIs --> MLs, Line 387 Kow (line 403?), Line 648 time spam? --> time span.

Answer. All these amendments have been made.

  • 7. Our opinion -->It is not suitable for these scientific articles. Out suggestion?

Answer: The heading of the paragraph has been amended into: Final considerations and suggestions